# Effect of Soybean Isoflavones on Proliferation and Related Gene Expression of Sow Mammary Gland Cells In Vitro

**DOI:** 10.3390/ani12233241

**Published:** 2022-11-22

**Authors:** Xinyan Ma, Yiyan Cui, Zhimei Tian, Miao Yu

**Affiliations:** Institute of Animal Science, Guangdong Academy of Agricultural Sciences, Key Laboratory of Animal Nutrition and Feed Science in South China, Ministry of Agriculture, State Key Laboratory of Livestock and Poultry Breeding, Guangzhou, Guangdong Public Laboratory of Animal Breeding and Nutrition, Guangzhou, Guangdong Engineering Technology Research Center of Animal Meat Quality and Safety Control and Evaluation, Guangzhou 510640, China

**Keywords:** soybean isoflavones, sow mammary gland cells, cell proliferation, gene expression

## Abstract

**Simple Summary:**

Isoflavones (ISO) improved the milk quality of sow, but no reports are available to elucidate the signaling mechanisms. To test the possible mechanisms, we investigated the effect of soybean ISO on the proliferation and related gene expression of sow mammary gland cells in vitro. The results showed that the cell proliferation and related gene or protein expression were enhanced by soybean ISO supplementation. Our findings indicated that ISO could promote the proliferation of sow mammary gland cells, which might be an additive to improve the lactation capacity and lactation performance of sows.

**Abstract:**

The present study was conducted to investigate the effects of synthetic soybean isoflavones (ISO) on the proliferation and related gene expression of sow mammary gland cells. Cells were cultured with 0 (control), 10, 20, or 30 μM of ISO under incubation conditions. After a 48 h incubation, these ISO-incubated cells proliferated more (*p* < 0.05) than the control cells. Cyclin E expression was higher (*p* < 0.05) in the 10 μM ISO and 20 μM ISO treatment groups than in the control group. Cyclin D1 and p21 expressions decreased (*p* < 0.05) with the 10 μM ISO treatment for 48 h. The relative mRNA abundances of the cells’ IG-1R (Insulin-like growth factor-1R), EGFR (Epidermal growth factor receptor), STAT3 (Signal transducer and activator of transcription 3) and AKT (protein kinase B) were enhanced (*p* < 0.05) by the 20 μM ISO treatment for 24 h and 48 h in the medium. The relative mRNA abundances of κ-casein at 48 h of incubation and β-casein at 24 h and 48 h of incubation were increased (*p* < 0.05) by 10 μM of ISO supplementation. It was concluded that ISO improved the proliferation of sow mammary gland cells, possibly by regulating cyclins and function genes expression in the cell proliferation signaling pathway.

## 1. Introduction

A large litter size in highly prolific sows is associated with a decrease in mean birth weight, and consequently, the vitality and growth of piglets, which is mainly attributed to the inadequate lactation of sows [1,2]. The proliferation of mammary epithelial cells plays a key role in improving the lactation performance of sows. During lactation, the decline of milk production occurred with a gradual decrease in the number of mammary epithelial cells within the mammary glands [3]. Mammary epithelial cell proliferation promotes rapid mammary gland development in sows, which improves their lactation performance. Studies have shown that plant extractions—such as isoflavones (ISO), with daidzin and genistin being the majority components—can improve cell proliferation [4,5]. It has been reported that ISO has anti-inflammatory, antioxidant, antiviral, and intestinal health improving properties [5,6,7,8]. Hu et al. [1] found that dietary glycitein promoted breast cell proliferation and increased milk yield and the protein content in sow milk, as well as enhanced the growth performance of the suckling piglets. Li et al. [9] reported that soy isoflavones improved the milk quality of sow by increasing serum antioxidant enzyme levels, thus, scavenging the free radicals in vivo.

A series of studies have been conducted to investigate how isoflavones affect mammary gland cell proliferation. However, these were evaluated in the breast cells of rat or human, and few were carried out by using sow mammary gland cells. Soy isoflavonoids have structural similarities to mammalian estrogens, bind and transactivate estrogen receptors, and induce proliferation in estrogen-sensitive breast tumor cells in culture [10,11]. Murrill et al. [12] found that daidzein promoted the proliferation of rat breast cells. Qin et al. [13] showed that soybean isoflavones promoted the breast cell proliferation of healthy premenopausal women and enhanced κ-casein mRNA expression, which helped to explain why the soybean isoflavone promoted milk protein synthesis. Mahn et al. [14] showed that soybean isoflavone could improve endothelial function and reduce blood pressure. Joy et al. [15] reported that a low concentration of soybean isoflavones stimulated ERK1/2 and AKT phosphorylation in human endothelial cells. Milk protein transcription was related to expressions of JAK/STAT pathway genes, such as transcription factor 3 (STAT3) and protein tyrosine kinase 2 (JAK2) [16]; the study reported that genistein suppressed the JAK2 and STAT3 expression, leading to proliferation inhibition in esophageal carcinoma cells, which suggested that increasing JAK2 and STAT3 expression could promote cell proliferation. Based on the aforementioned findings, we hypothesized that ISO might improve sow mammary gland cell proliferation through the PI3K-MAPK-AKT or JAK-STAT pathway observed in mammary cells. However, no reports are available to elucidate the signaling mechanisms of ISO in enhancing sow mammary gland cell proliferation. To test the above-mentioned hypothesis, we investigate the effect of soybean ISO on the proliferation and related gene expression of sow mammary gland cells in vitro.

## 2. Materials and Methods

### 2.1. Cell and Chemicals

Sow mammary cell line was cultured according to the protocol provided by the Department of Animal Science, Texas A&M University, College Station, TX, USA, which also offered the cell line to our lab. ISO (*p* ≥ 98%) was bought from Guangdong New Land Co. Dimethylsulfoxide (DMSO), 3-[4,5-dimethylthiazol-2-yl]-2,5- diphenyltetrazo- liumbromide (MTT), and Dulbecco’s modified Eagle’s Ham/F12 medium (DMEM/F12) were purchased from Sigma Chemical Co. (St. Louis, MO, USA). The antibodies of cyclin D1, p21, and cyclin E were Rabbit IgG (#2922, 1:200, Cell Signaling); Rabbit IgG (#sc-471, 1:100, Santa Cruz, CA, USA); and Mouse IgG (#4129S, 1:200, Cell Signaling Technology, Danvers, MA, USA), respectively. The second antibodies were Goat anti-Mouse IgG (H+L) Highly Cross-Adsorbed Secondary Antibody (Alexa Fluor 647, #A-21236, 1:200, ThermoFisher, Waltham, MA, USA); Goat anti-Rabbit IgG (H+L) Highly Cross-Adsorbed Secondary Antibody (Alexa Fluor 488, #A-11034, ThermoFisher).

### 2.2. Cell Culture and Treatment

The cells were seeded with 6 × 10^3^ cells per well in 96-well plates using growth medium (DMEM/F12 with 10% FBS, 5 μg/mL of insulin, 1 μg/mL of hydrocortisone, 5 ng/mL of EGF, and 1× PS Nantifungal/antibiotics). The ISO was diluted by DMSO stock solution, treatment media containing 10, 20, and 30 μM ISO with the same final content of 0.025% (*v*/*v*) DMSO. The cells were cultured at 37 °C under 5% CO_2_ and the media were changed every 2 days. Cell morphologies were observed and then collected after the 24 h or 48 h treatment for subsequent experimentation. Immunocytochemistry was detected at 48 h.

### 2.3. Cell Morphology and Cell Viability

ISO-induced morphological changes in the cells were observed using phase-contrast microscopy after treatment for 48 h (Axiovert 25 HBO 50/AC, Carl Zeiss, Jena, Gemany). Mammary cells were cultured in 96-well plates at 37 °C and exposed to varying concentrations (10, 20, 30 μM, *n* = 3, each) of ISO for 1–6 days. After treatment for 48 h, the quantities of cells in the 96-well plates were estimated by an MTT assay.

### 2.4. Immunocytochemistry

Immunocytochemistry was used to identify sow mammary cells and detect the protein of cyclin D1, p21, and cyclin E related to the cell cycle course. The sow mammary cells were identified using the immunocytochemistry method. The cells that were cultured in 6-well plates with sterile coverslips were fixed with 4% paraformaldehyde for 30 min at room temperature and washed with PBS for 3 times. The coverslips were incubated with 10% blocking bovine serum albumin in PBS for 30 min to suppress the non-specific binding of IgG. The coverslips were washed with PBS and incubated with primary antibody for 1 h. After rinsing 3 times with PBS, the coverslips were incubated for 30 min with fluorochrome-conjugated secondary antibody diluted 1:1000 with 3% bovine serum albumin in PBS for 30 min at 4 °C. Then, the cells were observed microscopically after rinsing 3 times with PBS and mounting with aqueous mounting medium. The optimal dilution of primary antibodies was determined as follows: casein (1:200), cyclin D1 (1:350), p21 (1:250), and cyclin E (1:400)

### 2.5. Functional Gene Expression of Mammary Epithelial Cells in Sows

Functional genes expressions were determined by a quantitative real-time PCR (qRT-PCR) [17]. The primers of the functional genes are listed in Table 1, and they were synthetized by Sangon Technical Co. Ltd. (Shanghai, China).

### 2.6. Western Blotting

The detailed operation methods followed the previous literature report [18]. Briefly, 100 mg of frozen tissue was homogenized with 1 mL of RIPA Lysis Buffer (Beyotime Institute of Biotechnology, Shanghai, China) and centrifuged (16,000× *g* for 15 min at 4 °C) to collect the supernatants. The protein concentration in the supernatant fluids was determined using the bicinchoninic acid (BCA) assay (Pierce, Rockford, IL, USA). All the samples were adjusted to equal protein concentrations and then diluted with 6× loading buffer, followed by denaturation in boiling water for 5 min. The denatured proteins were separated on 10% SDS-PAGE gel and transferred to a polyvinylidene difluoride (PVDF) membrane using the Bio-Rad transblot apparatus. Immunoblots were blocked with 3% BSA in Tris-Tween-buffered saline for 1 h at room temperature, and then incubated with primary antibodies diluted 1:1000 overnight at 4 °C with gentle shaking. The membranes were then incubated with the secondary antibody diluted 1:5000 for 1 h at room temperature. Chemiluminescence signals were detected using the ECL Plus TM Western Blotting Substrate (Thermo Fisher Scientific, Waltham, MA, USA), according to the manufacturer’s instructions, and performed on a ChemiDoc XRS imaging system (Bio-Rad, Hercules, CA, USA). The intensity of the bands was analyzed using QuantityOne software (Bio-Rad, USA), and the results were expressed as the abundance of each target protein relative to β-actin.

### 2.7. Statistical Analyses

The effects of ISO at the various concentrations were analyzed by a one-way analysis of variance using computing software SAS (v6.12, SAS Institute, Cary, NC, USA). Data were presented as means ± SD (standard deviation). The replicate served as the experimental unit. The least significant difference (LSD) method was used to compare the differences between treatment means. Differences were considered statistically significant at *p* < 0.05.

## 3. Results

### 3.1. Identification of Sow Mammary Cell

Primary sow mammary cells were identified using keratin antibody, which is a marker protein for mammary cells. Keratin protein existed in the sow mammary gland cells (Figure 1), which indicated that they were the target cells needed in the experiment.

### 3.2. Effects of ISO on Proliferation of Sow Mammary Cells

Cells in the presence of ISO were lost less compared with that of the control, and the cells cultured with 10 μM or 20 μM of ISO grew better (Figure 2). The OD values represented the cell proliferation (Figure 3). ISO at 10 μM, 20 μM and 30 μM increased (*p* < 0.05) the cell proliferation at day 2. ISO at 10 μM and 20 μM increased (*p* < 0.05) the cell proliferation and 30 μM of ISO decreased (*p* < 0.05) it at day 5, respectively. ISO at 10 μM increased (*p* < 0.05) the cell proliferation and 30 μM of ISO decreased (*p* < 0.05) it at day 6, respectively. The cell proliferation was higher (*p* < 0.05) in the 10 μM ISO treatment than in all other ISO treatments on day 2, day 5, or day 6.

### 3.3. Identification of Key Proteins Related to Cell Proliferation

The result showed that cyclin D1 and cyclin E were enhanced by the ISO that was added, while p21 was weakened with the 10, 20, or 30 μM ISO treatment for 48 h (Figure 4). The protein abundance of cyclin D1 and p21 was decreased (*p* < 0.05) with the 10 μM ISO treatment for 48 h (Figure 5). Compared to the control, the protein abundance of cyclin E was increased (*p* < 0.05) by the 10 μM ISO supplementation for 48 h and increased (*p* < 0.05) by the 20 μM ISO supplementation for 48 h. Compared to the control, the protein abundance of cyclin E was increased (*p* < 0.05) by the 10 μM, 20 μM and 30 μM ISO supplementation for 48 h.

### 3.4. Functional Gene Expression in Mammary Gland Cell

IGF-1R mRNA expression was increased (*p* < 0.05) by the 10 μM ISO supplementation at 24 h, and by the 20 μM and 30 μM ISO supplementation at 48 h, respectively (Figure 6A). EGFR mRNA expression was increased (*p* < 0.05) by the 20 μM ISO supplementation at 24 h and 30 μM supplementation at 48 h, respectively (Figure 6B). STAT3 and AKT mRNA expression were increased (*p* < 0.05) by the 20 μM ISO supplementation at 24 h and 48 h (Figure 6C, D). β-casein mRNA expression was increased (*p* < 0.05) by the 10 μM ISO supplementation at 24 h and 48 h (Figure 6E). κ-casein mRNA expression was increased (*p* < 0.05) by the 10 μM ISO supplementation at 48 h (Figure 6F).

## 4. Discussion

Our data demonstrated that ISO improved the proliferation of sow mammary gland cells by regulating the cyclins expression or other related function genes expression of the PI3K-MAPK-AKT or JAK-STAT pathway, which supports our hypothesis. Previous studies have shown that genistein, at concentrations of 0.1~10 μM, stimulates the proliferation of human breast cancer cells in culture [19,20]. Gaete et al. [21] reported that daidzein stimulated the growth of breast cancer cells and potentiated estrogen-induced cell proliferation in the uterus. In our study, we found that sow mammary gland cell proliferation increased significantly with ISO supplement compared with the control group, and it showed time and dose dependence. Our results agree with previous studies, suggesting that ISO can improve cell viability. These data provide references for understanding the sow mammary gland cell proliferation regulated by ISO in vitro.

Studies on the mechanism of cell cycle regulation showed that G1 phase was the main regulatory point of cell proliferation [22]. Crossing the checkpoints of G1/S phase, the progress of the cell cycle will become irreversible [23]. An active cyclin-CDK promoted cyclin E transcription, which would drive cell transformation from G1 phase to S phase and induce cell growth and proliferation [24,25]. In this experiment, the expression of cyclin E mRNA and protein content in the ISO-treated group was significantly higher than that in the control group, which indicated that ISO could promote the expression of cyclin E and, finally, enhance cell proliferation. The expression of cyclin D1 was cell cycle dependent, with the highest expression in G0/G1 phase and the lowest expression in S and G2/M phases [23]. The expression level of cyclin D1 in the ISO treatment group was lower than that in the control group, suggesting that the cells were in different cell cycles in each group. Possibly, the cells were in S phase or G0/G1 phase in the ISO treatment groups or control, respectively. Cyclins bound to CDK inhibitors (CDKIs) and inhibited cell proliferation [26]. p21, as a kind of CDKI, was increased in senescent cells [27]. In this study, p21 expression in the ISO treatment groups was significantly lower than that in the control group, indicating that ISO could fight against cell senescence and promote cell proliferation.

There are few reports about the effect of ISO on the functional genes of sow mammary epithelial cells in vitro. We found that ISO promoted the expression of functional genes in sow mammary epithelial cells. IGF-1R, EGFR, STAT3, and AKT are key genes of breast cell proliferation, in which mRNA levels were increased by ISO supplement. It is reported that IGF-1R regulated cell mitosis and promoted cell proliferation and differentiation [28]. The combination of IGF-1/IGF-1R promoted cell proliferation and differentiation [29]. EGFR, a kind of transmembrane glycoprotein with multiple functions and tyrosine kinase activity, widely exists on the surface of various mammalian tissues and cells [30]. With tyrosine kinase activity activated, EGFR stimulates downstream signals to mediate cell proliferation, differentiation, survival, and other processes [31]. Therefore, it is speculated that ISO promotes cell proliferation by increasing EGFR mRNA expression to activate tyrosine kinase activity. STAT3 acts on the cell cycle and promotes cells to enter S phase from G1 [32]. Studies have shown that STAT3 can induce a high expression of key genes, closely relating to the promotion of cell proliferation and differentiation and inhibiting cell apoptosis [33,34,35]. In the present study, IGF-1R, EGFR, and STAT3 mRNA in the added ISO were significantly higher than the control group, which further indicated that ISO could fight against breast epithelial cells apoptosis and promote cell proliferation and differentiation. The results were consistent with previous studies showing that IGF-1R, EGFR, and STAT3, which act as signal factors, promote cell proliferation through JAK/STAT signal pathway genes [16].

AKT is another important signal factor for cell proliferation, which promotes the phosphorylation of downstream glycogen synthase kinase-3 β (GSK-3β) to elevate cell proliferation through the PI3K/AKT signaling pathway [36,37]. In this study, AKT mRNA expression increased with the 20 μM ISO supplementation, suggesting that ISO stimulated AKT mRNA expression and promoted cell proliferation through the PI3K/AKT signaling pathway, which is in accordance with the report of Joy et al. [15].

Casein is a key index by which evaluate the quality of milk. In milk, 80% of protein is casein, more than 36% of which is β-casein [38]. κ-casein (κ-CN) plays an important role in maintaining the stability of casein micelles, which has a significant impact on milk quality [39]. Qin et al. [13] demonstrated that soybean ISO promoted breast cell proliferation and enhanced κ-casein mRNA expression. In our study, ISO increased the mRNA levels of κ-casein and β-casein, which is consistent with previous reports. It was reported that milk protein synthesis was mainly regulated by the JAK-STAT signaling pathway [40,41]. Zhou et al. [42] found that the STAT5 gene of bovine mammary epithelial cells promoted the expression of β-casein mRNA. Santos et al. [43] reported that estrogen promoted the expression of STAT5 through binding with its receptor and regulated the transcription of β-casein gene in breast epithelial cells [44]. Estrogen directly regulates the transcription and translation of milk protein genes in the breast and affects milk protein synthesis and secretion [45]. Just like estrogen, ISO (which is estrogen-like)may up-regulate β-casein and κ-casein mRNA expression by activating the JAK2-STAT5 signaling pathway, which has been further verified by previous reports [13,15].

## 5. Conclusions

The results suggested that soybean ISO improved cell proliferation by regulating the expression of cyclin proteins and the related functional gene expression of sow mammary gland cells. The optimized dose of soybean ISO for stimulating mammary epithelial cells proliferation was 20 μM.

## Figures and Tables

**Figure 1 animals-12-03241-f001:**

Identification of sow mammary gland cells. Immunocytochemistry was used to identify the cells. The magnification scale was 400×.

**Figure 2 animals-12-03241-f002:**
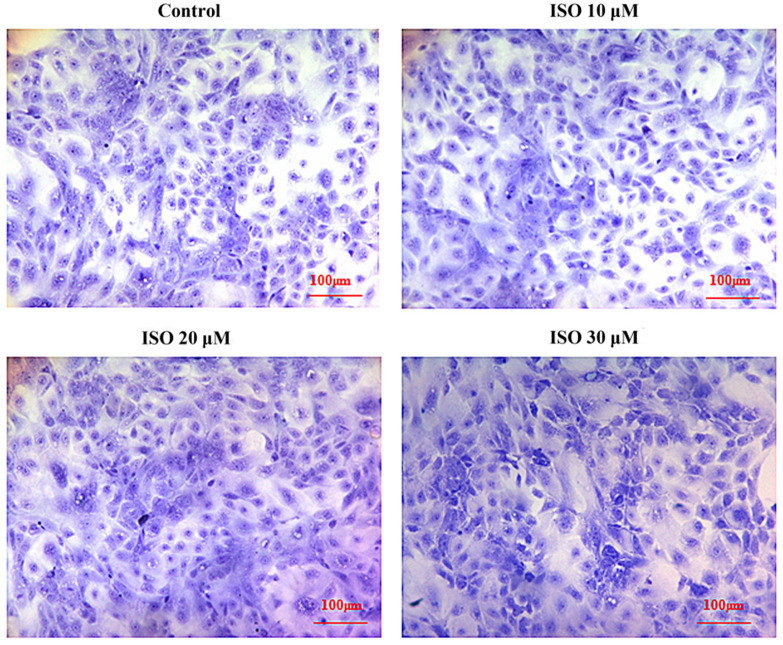
Effects of ISO on morphology of sow mammary gland cells. The morphology of the cell under different concentrations of ISO.

**Figure 3 animals-12-03241-f003:**
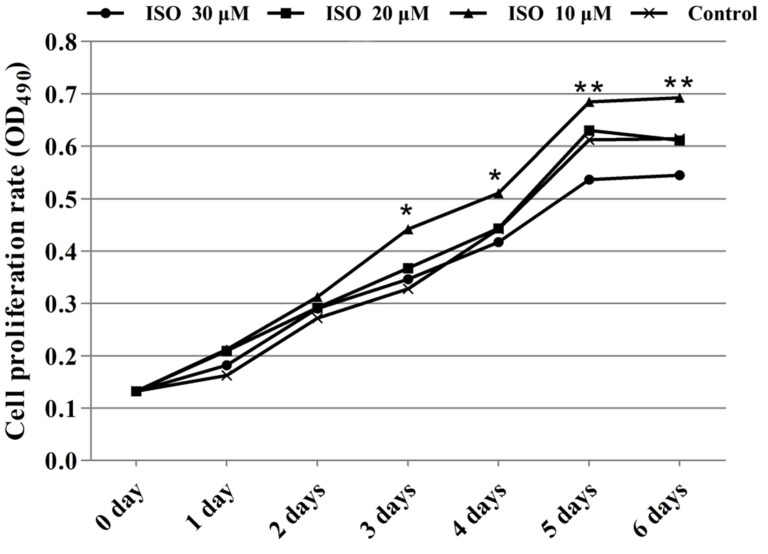
Effects of ISO on cell proliferation from day 0 to day 6. Groups indicated by * (*p* < 0.05) or ** (*p* < 0.01) differ significantly compared to control.

**Figure 4 animals-12-03241-f004:**
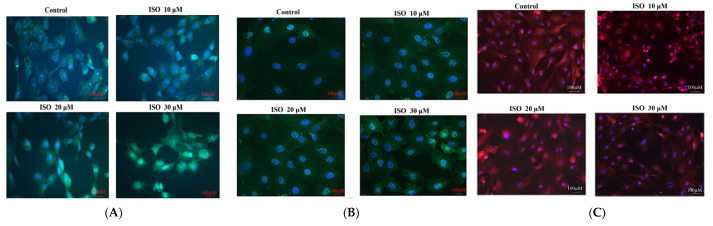
Effects of ISO on cyclins. Immunofluorescence analysis was positive for cyclins with different ISO concentration supplementations. (**A**) Cyclin D1 protein; (**B**) cyclin E protein; (**C**) p21 protein.

**Figure 5 animals-12-03241-f005:**
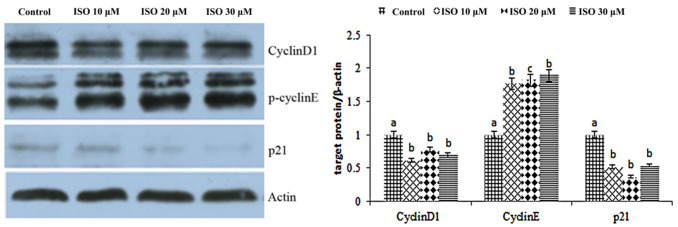
Effects of ISO on cyclins protein expression. Western blot was used to analyze cydin D1, cydin E, and p21 protein. Each value was mean ± SD (*n* = 3). Different letters on the histogram indicate significant differences (*p* < 0.5) in the mRNA expression of each group.

**Figure 6 animals-12-03241-f006:**
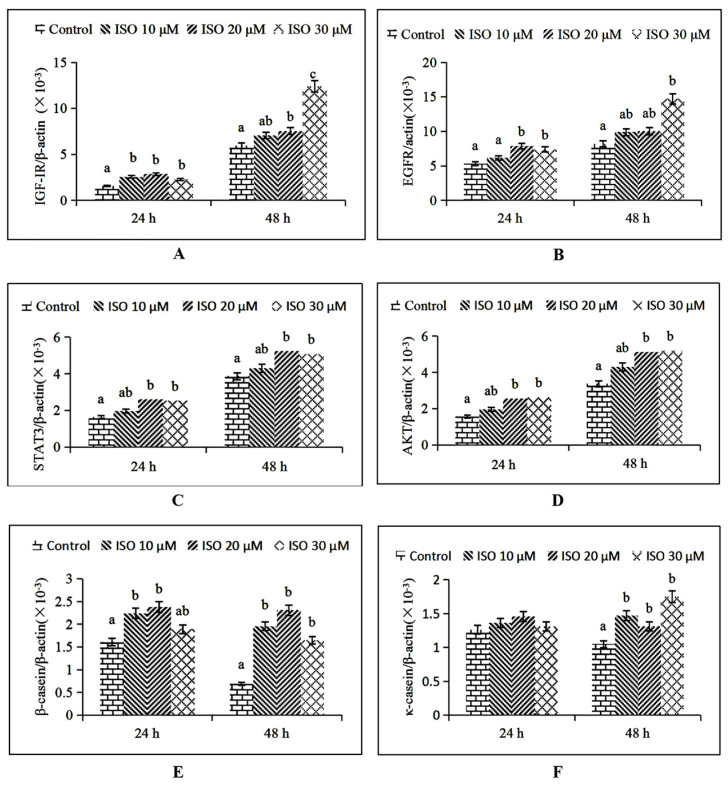
Effects of ISO on key genes expression related to cell function. After ISO added to the cells, cells were then collected at different time points (24 and 48 h), and the mRNA expression levels of the IGF-1R, EGFR, STAT3, AKT, β-casein, and κ- casein genes were measured by RT-qPCR (**A**–**F**). The chicken β-actin gene was used as the reference gene. Different letters on the histogram indicate significant differences (*p* < 0.05) in the mRNA expression of each group. Each value was mean ± SD (*n* = 3). IGF-1R = Insulin-like growth factor-1R; EGFR = Epidermal growth factor receptor; STAT3 = Signal transducer and activator of transcription 3; AKT = protein kinase B.

**Table 1 animals-12-03241-t001:** Primers of genes used for qRT-PCR.

Gene Name	Forward Primer (5′-3′)	Reverse Primer (3′-5′)	Accession Numbers	Product Length (bp)
β-actin	CACGCCATCCTgCGTCTGGA	AGCACCGTGTTGGCGTAGAG	DQ452569.1	270
IGFR	CTCCAAGCCTAAGCAAAATGAT	TGCGTGGTGAAGACTCCGTC	NM_001291858.2	259
EGFR	GGATAGGGATTGGCGAGTTT	GCATAGCACAGGTTTCGGTTT	NM_214007.1	400
STAT3	TGGGTGGAGAAGGACATCA	TAGACCAGCGGAGACACAAG	HM462247.1	149
AKT	CCTGAAGAAGGAGGTCATCG	TCGTGGGTCTGGAAGGAGTA	NM_001159776.1	123
β-casein	CTTGATCGCCATGAAGCTC	GAGCAGAGGCAGAGAAAGGAC	EU213063.1	472
k-casein	GACGCTGGACTTCCTTCGAGATC	CAGAAAAGACACAGTCCAAGGCG	X51977.1	196

## Data Availability

The raw datasets used and analyzed during the current study are available from the corresponding author on reasonable request.

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
