# Peer review of "Effect of Soybean Isoflavones on Proliferation and Related Gene Expression of Sow Mammary Gland Cells In Vitro"

_animals, 2022, doi:10.3390/ani12233241_

Round 1

Reviewer 1 Report

General comments:

    This study provided some new information on these aspects studied. The experiment was well desinged and performed. The methods used are appropriate. The manuscript was well organized. I recommend its publication after some minor revisions.

Specific comments:

1) L 12: add "are" between reports and available.

2) L 13: change "possibility" into "possible", and "investigate" into "investigated".

3) L 16: change "supplemented" into "supplementation".

4) L 22: change "proliferative" into "proliferated".

5) L 23: delete "those".

6) L 60: change "that were founded in mammality cells" into "observed in mammary cells ", and add "are" after "no reports".

7) L 61: change "enhance" into "in enhancing".

8) L 62: change "possibilty mentioned above" into "the above-mentioned hypothesis".

9) L 67: change "followed" into "according to".

10) L 126-131: change "at 2th" into "d 2", and check and revise those similar expressions throughout the manuscript, and add the description: The cell proliferation was higher (P<?) in the 10 uM ISO treatment than in all of other ISO treatments on d 2, d 5 or d 6.

11) L 171: after "pathway", add ", which has supported our hypothesis." .

12) L 237: change "improved" into "would improve".

13) L 240: change "20 uM" into "10 uM".

Author Response

General comments:

    This study provided some new information on these aspects studied. The experiment was well desinged and performed. The methods used are appropriate. The manuscript was well organized. I recommend its publication after some minor revisions.

Specific comments:

1) L 12: add "are" between reports and available.

Thank you for your comment.

In line12, available is an adjective, which was used to precede the noun (reports). While, reports and available were not coordinating relation. So, shall I add "are" between reports and available? Please tell me, thanks very much.

2) L 13: change "possibility" into "possible", and "investigate" into "investigated".

Thank you for your revision.

We have corrected in the article.

3) L 16: change "supplemented" into "supplementation".

Thank you for your revision.

In line16, "supplemented" has been changed to "supplementation".

4) L 22: change "proliferative" into "proliferated".

Thank you for your revision.

In line22, "proliferative" has been changed to "proliferated".

5) L 23: delete "those".

Thank you for your revision.

In line23, "those" has been deleted.

6) L 60: change "that were founded in mammality cells" into "observed in mammary cells ", and add "are" after "no reports".

Thank you for your revisions.

In line60, we have changed "that were founded in mammality cells" into "observed in mammary cells ", and added "are" after "no reports".

7) L 61: change "enhance" into "in enhancing".

Thank you for your revision.

 In line71, we have changed "enhance" into "in enhancing"

8) L 62: change "possibilty mentioned above" into "the above-mentioned hypothesis".

Thank you for your revision.

In line62, we have changed "possibilty mentioned above" into "the above-mentioned hypothesis".

9) L 67: change "followed" into "according to".

Thank you for your revision.

In line67, we have changed "followed" into "according to".

10) L 126-131: change "at 2th" into "d 2", and check and revise those similar expressions throughout the manuscript, and add the description: The cell proliferation was higher (P<?) in the 10 uM ISO treatment than in all of other ISO treatments on d 2, d 5 or d 6.

Thank you for your comments.

Among L126-131, we have changed the expression and added the related content just as your suggestion.

11) L 171: after "pathway", add ", which has supported our hypothesis." .

Thank you for your revision.

In line171, we have added ", which has supported our hypothesis." after "pathway".

12) L 237: change "improved" into "would improve".

Thank you for your revision.

In line237, "improved" has been changed into "would improve".

13) L 240: change "20 uM" into "10 uM".

Thank you for your comment. We felt so sorry to confuse you.

In the article, the results showed that adding 10 μM was beneficial to some indicators, but couldnot meet other indicators(eg. key proteins expression related to cell proliferation in figure 5 and functional gene expression in figure 5, ). With 20μM added, it could basically meet all the evaluation indicators used in the paper. So, it suggested that the optimized dose of soybean ISO for stimulating mammary epithelial cells proliferation was 20 μM.

We appreciate your efforts again.

Reviewer 2 Report

The authors investigated how synthetic soybean isoflavones affect gene expression in mammary cells in vitro.

The work is relevant, however, I have some concerns:
- STAT3 expression was quantified, however, the importance of this gene was not pointed out in the introduction, but STAT5 effects were emphasized, and also extensively in the discussion section. Why do you choose STAT3 rather than STAT3.

- My main concern is related to the cell cycles in which treatments were tested as the authors referred in lines 189-190 that "The expression level of CyclinD1 in the ISO treatment group was lower than that in the control group, suggesting cells were in different cell cycles of each group."

- Authors should review English.

- I don't understand the phrase "The combination of IGF-1 and IGF-1R promoted cell proliferation and differentiation [28]." (line 200). Maybe the author refers to IGF-1/IGF-IR binding? Could you clarify, please?

- In lines 223-224, the authors referred that κ-casein "has a significant impact on milk yield and quality [38]". Indeed it is extensively reported in the literature that k-casein affects milk quality and milk ability for cheesemaking, however, it does not influence directly milk yield, but its expression can be corrected with milk yield. The authors should rephrase, maybe based on additional references. 

Author Response

The authors investigated how synthetic soybean isoflavones affect gene expression in mammary cells in vitro.

The work is relevant, however, I have some concerns:
1)- STAT3 expression was quantified, however, the importance of this gene was not pointed out in the introduction, but STAT5 effects were emphasized, and also extensively in the discussion section. Why do you choose STAT3 rather than STAT3.

Thank you for your comments. Please let us explain as follow. Thanks, again.

STAT3 acts on the cell cycle and promotes cells to enter S phase from G1. Studies have shown that STAT3 can induce high expression of key genes, closely relating to promote cell proliferation and differentiation and inhibit cell apoptosis. Corroding to previous studies, we chose STAT3. In the introduction, STAT3 related reference was also supplemented.

2)- My main concern is related to the cell cycles in which treatments were tested as the authors referred in lines 189-190 that "The expression level of CyclinD1 in the ISO treatment group was lower than that in the control group, suggesting cells were in different cell cycles of each group."

Thank you for your comments.

It was just an explanation for the present result. We will study further to explore how the ISO affect the expression level of CyclinD1.

3)- Authors should review English.

Thanks for your suggestion. We apologize that our written English has bothered you. We will do our best to improve our written English. 

4)- I don't understand the phrase "The combination of IGF-1 and IGF-1R promoted cell proliferation and differentiation [28]." (line 200). Maybe the author refers to IGF-1/IGF-IR binding? Could you clarify, please?

Thank you for your comments. We felt so sorry to confuse you.

It mean IGF-1/IGF-IR binding. IGF-1 and IGF-1R were a pair gens, and IGF-1 combine to IGF-1R to promote cell proliferation and differentiation. We changed “IGF-1 and IGF-1R” into “IGF-1/IGF-IR” in line 200.

5)- In lines 223-224, the authors referred that κ-casein "has a significant impact on milk yield and quality [38]". Indeed it is extensively reported in the literature that k-casein affects milk quality and milk ability for cheesemaking, however, it does not influence directly milk yield, but its expression can be corrected with milk yield. The authors should rephrase, maybe based on additional references. 

Thank you for your comments.

The reference showed that κ - casein (κ - CN) plays an important role in maintaining the stability of casein micelles, which has a significant impact on milk quality. So, We rephrased this sentence and deleted “yield”.

We appreciate your efforts again.

Reviewer 3 Report

1. In line 36 of the manuscript, the author should elaborate on how the proliferation of mammary epithelial cells improves lactation performance of sows. This issue is too critical to be discussed in general.

2. In line 81-83 of the manuscript, is the content of DMSO always 0.025% when media with different concentrations of ISO content are configured? If the solubilization power of DMSO is certain, then the DMSO in the medium with different concentrations of ISO content will be different. Therefore, it is recommended that the authors make a table indicating the amount of DMSO in ISO medium at different concentrations. In addition, the authors should have informed the control group of DMSO levels.

3.It is suggested that the authors add a column in Table 1, and the NCBI accession numbers of all tested genes should be added in the table.

4.It is known from 3.2 that the authors examined the proliferation of mammary epithelial cells for 6 consecutive days, but why not explain this in Section 2.3? At the same time, the photographing time of MTT experiment should be supplemented.

5.The procedure of the Western blot experiment in 2.6 should be described briefly.

6. In the mammary epithelial cell identification experiment, it is generally necessary to put three maps, and only one merge map is put here. In addition, the authors need to label the magnification scale.

7.In Figure 3, the authors used bar graphs to show the effect of ISO concentrations on mammary epithelial cell proliferation. Why not use a line graph to show the effect, so that it would be clearer.

8. The font above the photo in Figure 4 is too small compared to the other pictures, and A, B, and C are in the wrong positions on the three pictures.

9. It is suggested that the authors use a uniform shape for all ISO groups with different concentrations in the bar charts in the manuscript, so as to increase the uniformity of the article.

10. In the pathway verification experiments, many pathway-related proteins reflect cell proliferation or apoptosis by the ratio of phosphorylated and non-phosphorylated. The authors used fluorescence quantitative experiments to prove this problem, and the results are debatable. In addition, in the verification pathway experiment, gene selection is less, so it is difficult to draw the conclusion from this manuscript.

Author Response

Comments and Suggestions for Authors.

  1. In line 36 of the manuscript, the author should elaborate on how the proliferation of mammary epithelial cells improves lactation performance of sows. This issue is too critical to be discussed in general.

Thank you for your comments.

We have supplemented the related content to explain how the proliferation of mammary epithelial cells improves lactation performance of sows.

  1. In line 81-83 of the manuscript, is the content of DMSO always 0.025% when media with different concentrations of ISO content are configured? If the solubilization power of DMSO is certain, then the DMSO in the medium with different concentrations of ISO content will be different. Therefore, it is recommended that the authors make a table indicating the amount of DMSO in ISO medium at different concentrations. In addition, the authors should have informed the control group of DMSO levels.

Thank you for your comments. We felt so sorry to confuse you.

The ISO was diluted with DMSO stock solution and added into the serum-free media to reach final concentrations of 10, 20 and 30 uM. Meanwhile, the final content of DMSO were adjusted to be 0.025% (v/v) in treatment media with DMSO stock solution for maintaining same DMSO content. So we revised the sentence “The ISO was diluted to 10, 20 and 30 μM by DMSO. The control cells were treated in same amount of DMSO (0.025% (v/v)) as treatment media.” into “The ISO was diluted by DMSO stock solution, treatment media contain 10, 20 and 30 μM ISO with the same final content of 0.025% (v/v) DMSO.”

3.It is suggested that the authors add a column in Table 1, and the NCBI accession numbers of all tested genes should be added in the table.

Thank you for your suggestions.

The accession numbers were supplemented in Table 1.

4.It is known from 3.2 that the authors examined the proliferation of mammary epithelial cells for 6 consecutive days, but why not explain this in Section 2.3? At the same time, the photographing time of MTT experiment should be supplemented.

Thank you for your comments.

The missing content were supplemented in Section 2.3.

The photographing time of MTT experiment was the time that cells were treated with ISO for 48 h.

5.The procedure of the Western blot experiment in 2.6 should be described briefly.

Thank you for your comment.

The procedure of the western blot was supplemented in section 2.6.

  1. In the mammary epithelial cell identification experiment, it is generally necessary to put three maps, and only one merge map is put here. In addition, the authors need to label the magnification scale.

Thank you for your comments.

Three maps were put in Figure 1 and the magnification scale was labeled.

7.In Figure 3, the authors used bar graphs to show the effect of ISO concentrations on mammary epithelial cell proliferation. Why not use a line graph to show the effect, so that it would be clearer.

Thank you for your suggestions.

We afraid that too much data (seven time points and four treatments) made it difficult to distinguish in a line graph, so we used bar graphs. However, we drew a line graph to show the effect (as follow). Please compare the bar graph and line graph to use a better one.

We appreciate your efforts again.

Fig.3 Effects of ISO on cell proliferation from 0d to 6d. Groups indicated by

* (P<0.05) or ** (P<0.01) differ significantly compared to control.

  1. The font above the photo in Figure 4 is too small compared to the other pictures, and A, B, and C are in the wrong positions on the three pictures.

Thank you for your comments.

They were corrected in Figure 4.

  1. It is suggested that the authors use a uniform shape for all ISO groups with different concentrations in the bar charts in the manuscript, so as to increase the uniformity of the article.

Thank you for your suggestion. In order to distinguish different indicators, we use different shape for the different ISO groups. It’s appreciated for us to be understood.

We appreciate your efforts again.

  1. In the pathway verification experiments, many pathway-related proteins reflect cell proliferation or apoptosis by the ratio of phosphorylated and non-phosphorylated. The authors used fluorescence quantitative experiments to prove this problem, and the results are debatable. In addition, in the verification pathway experiment, gene selection is less, so it is difficult to draw the conclusion from this manuscript.

 Thank you for your comments. In our further study, we will evaluate more indices, such as enzymatic activity, relevant gene expression and protein expression in the signal pathway an so on, to reflect cell proliferation or apoptosis. Meanwhile, we expect that the professor will give us more suggestions in our further study. Thank you again for your comments.

Round 2

Reviewer 3 Report

In the modification of Figure 1, the author does not understand what I mean. The three figures refer to the binding of the antibody to the target protein, the DAPI staining of the nucleus, and the last one is the merge of the previous two.

Author Response

Dear referee,
    Thanks for your comments. We have carefully checked the full text again, and made corresponding adjustments for the written English. Meanwhile, we used “Track Changes” function.

   In Figure 1, the two figures refer to the binding of the antibody to the target protein (keratin) and the DAPI staining of the nucleus (DAPI) were supplemented.

    We appreciate your efforts again.

Best wishes.

Yours sincerely,

                                                      Xinyan MA, Yiyan CUI, Zhimei TIAN, Miao YU
